# Computational Modelling of Tone Perception Based on Direct Processing of *f*_0_ Contours

**DOI:** 10.3390/brainsci12030337

**Published:** 2022-03-02

**Authors:** Yue Chen, Yingming Gao, Yi Xu

**Affiliations:** 1Department of Speech, Hearing and Phonetic Sciences, University College London, London WC1N 1PF, UK; yue.chen.1@ucl.ac.uk; 2Institute of Acoustics and Speech Communication, TU Dresden, 01069 Dresden, Germany; yingming.gao@mailbox.tu-dresden.de

**Keywords:** speech perception, Mandarin tones, tone recognition, tone features

## Abstract

It has been widely assumed that in speech perception it is imperative to first detect a set of distinctive properties or features and then use them to recognize phonetic units like consonants, vowels, and tones. Those features can be auditory cues or articulatory gestures, or a combination of both. There have been no clear demonstrations of how exactly such a two-phase process would work in the perception of continuous speech, however. Here we used computational modelling to explore whether it is possible to recognize phonetic categories from syllable-sized continuous acoustic signals of connected speech without intermediate featural representations. We used Support Vector Machine (SVM) and Self-organizing Map (SOM) to simulate tone perception in Mandarin, by either directly processing *f*_0_ trajectories, or extracting various tonal features. The results show that direct tone recognition not only yields better performance than any of the feature extraction schemes, but also requires less computational power. These results suggest that prior extraction of features is unlikely the operational mechanism of speech perception.

## 1. Introduction

How exactly speech perception works is still a mystery. It is widely assumed that multiple acoustic cues are needed for the perception of segments (consonants and vowels) and suprasegmentals (tone, intonation, etc.), and a major goal of research is to find out which cues are relevant for the recognition of these units [1,2]. For example, formants may provide primary cues for signaling different vowel categories, VOT is useful for distinguishing between voiced and voiceless plosives [3], pitch contour and pitch height are useful for differentiating lexical tones [4,5], etc. During speech perception, those cues are detected and then combined to identify specific contrastive phonetic units such as consonants, vowels, or tones [6]. This assumed mechanism, therefore, consists of two phases: feature detection, and phonetic recognition. No research so far, however, has demonstrated how exactly such a two-phase process can achieve the recognition of phonetic units in the perception of continuous speech. At the same time, another possibility about speech perception has rarely been theoretically contemplated, namely, a mechanism in which raw acoustic signals are processed to directly recognize phonetic units, without the extraction of intermediate featural representations. It may be difficult to test this hypothesis with existing accounts, however, because conventional behavioral and neural studies can only allow us to explore what acoustic cues are important for perception, but not how the whole perception process may work. What is needed is a way to explore the operation of speech perception by simulating it as a step-by-step procedural process, starting from acoustic signals as input, and ending with identified phonetic categories as output. This can be done through computational modeling that implements each proposed perception model as a phonetic recognition system. The present study is an attempt to apply this paradigm by making a computational comparison of direct phonetic perception to various two-phase perception models. In order to avoid the pitfall, often seen in computational modeling, of allowing multiple hidden layers that do not correspond to specific aspects of theoretical models of perception, here we try to construct computational models with transparent components that correspond explicitly to specific aspects of the related theoretical models.

### 1.1. Feature-to-Percept Theories

#### 1.1.1. Distinctive Feature Theories

A major source of the feature-based view of speech perception is the classic theory of distinctive features [7]. The theory was proposed as an attempt to economize the representation of speech sounds beyond segmental phonemes [8,9]. In a pursuit to identify the most rudimentary phonetic entities, an even smaller set of featural contrasts than phonemes, aka distinctive features, was proposed [10]. Jakobson et al. [7] proposed a system with only 12 pairs of features, each making a binary contrast based predominantly on acoustic properties. An alternative system was proposed by Chomsky and Halle [11] with a much larger set of binary features (around 40) that are predominantly based on articulatory properties. Some phonological theories have even claimed that distinctive features are the true minimal constituents of language [12,13]. Most relevant for the current discussion, it is often assumed that the detection of the discrete features [14,15], be it binary or multivalued [1], is the key to speech perception [16,17]. This is seen in both the auditory theories and motor theories of speech perception, two competing lines of theories that have been dominating this area of research, as will be outlined next.

#### 1.1.2. Auditory Theories

Auditory theories as a group assume that perceptual cues of phonetic contrasts are directly present in the acoustic signals of speech [17,18,19,20]. These theories assume that it is the distinctive acoustic properties that listeners are primarily sensitive to, and that speech perception is achieved by either capturing these properties [21] or extracting distinctive features [22]. These theories are often presented in opposition to motor theory, to be discussed next, in that they assume no intermediate gestural representations between acoustic cues and perceived categories. They recognize a role of distinctive features, and assume a need for extracting them in perception [17]. This need is elaborated in the Quantal Theory [23,24,25,26] based on the observation that auditory properties are not linearly related to continuous changes of articulation, but show some stable plateau-like regions in the spectrum. The plateaus are argued to form the basis of universal distinctive features. In addition, it is further proposed that there are enhancing features to augment the distinctive features [15,18,26,27,28].

#### 1.1.3. Motor Theories

The motor theory [29,30,31], in contrast, assumes that the peripheral auditory processing phase of speech perception is followed by an articulatory recognition phase, in which articulatory gestures such as tongue backing, lip rounding, and jaw raising are identified. The motor theory is mainly motivated by the observation of the lack of one-to-one relations between acoustic patterns and speech sounds [32,33]. It is argued that invariance must lie in the articulatory gestures that generate the highly variable acoustic patterns. Therefore, gestures would serve as intermediate features that can match the auditory signals on the one hand, and the perceived phonemic or lexical units on the other hand.

Counter evidence to the motor theory comes from findings that speech perception can be achieved without speech motor ability in infants [34], non-human animals [35], and people suffering from aphasia [36,37,38]. However, there is also increasing evidence that perceiving speech involves neural activity of the motor system [39,40], and the motor regions are recruited during listening [41,42,43,44]. A range of brain studies using methods like TMS also showed evidence for the motor theory [45,46,47].

However, perceptual involvement of the motor system is not direct evidence for gesture recognition as the necessary prerequisite to phonetic recognition. For one thing, it is not clear whether motor activations occur before or after the recognition of the perceived categories. For another thing, there is increasing evidence that motor area activation mostly occurs only under adverse auditory conditions [45,48,49], which means that motor involvement may not be obligatory for normal perception tasks.

An issue that has rarely been pointed out by critics of motor theory is that gestures are actually not likely to be as invariant as the theory assumes. While a consonantal gesture could be in most cases moving toward a constricted vocal tract configuration, a vowel gesture could be either a closing or opening movement, depending on the openness of the preceding segment. In this respect, a greater degree of articulatory invariance is more likely found in the underlying phonetic targets in terms of vocal tract and/or laryngeal configuration rather than the target approximation movements [50,51,52].

### 1.2. Feature-to-Percept vs. Direct Phonetic Perception

As mentioned above, what originally motivated the motor theory, which has also permeated much of the debate between the motor and auditory theories is the apparent and pervasive variability in the speech signal. This is an issue fundamental for any theory of speech perception, namely, how is it possible that speech perception can successfully recover the phonetic categories intended by the speaker despite the variability? Note that, however, the question can be asked in a different way. That is, despite the variability, is there still enough within-category consistency in the acoustic signal that makes speech perception effective? If the answer is yes, the next question would be, what is the best way to capture the within-category consistency?

The answer by all the theories reviewed above would be that a feature-based two-phase process is the best way to capture the within-category consistency. They differ from each other only in terms of whether the extracted features are primarily auditory or articulatory, or a mixture of both as in the case of distinctive features. This commonality is nicely illustrated in Figure 1 from Fant [53]. Here, after being received by the ear, and having gone through the primary auditory analysis, the acoustic signals of speech are first turned into featural patterns that are either auditory (intervals CD) or motor (interval GF). Either way, they are both sub-phonemic and featural, and need to be further processed to identify the categorical phonemes, syllables, words, etc.

A key to this two-phase concept is the assumption that for each specific phonetic element only certain aspects of the speech signal are the most relevant, and what needs to be theoretically determined is whether the critical aspects are auditory or motor in nature. This implies that the non-critical properties (i.e., those beyond even the enhancing features) are redundant and are therefore not taken into account in perception. Even though there is also recognition that certain minor cues can be useful [54], it is generally felt that the minor cues are not nearly as important. There is little discussion, however, as to how the perception system can learn what cues to focus on and what cues to ignore.

An alternative possibility, as explored in the present study, is that raw acoustic signals of connected speech, after an initial segmentation into syllable sized chunks, can be processed as a whole to directly recognize the relevant phonetic elements, such as consonants, vowels, and tones. This process does not consist of a phase in which auditory or articulatory features are explicitly identified and then used as input to the recognition of the units at the phonemic level. There are a number of reasons why such a direct perception of phonetic categories may be effective. First, given that speakers differ extensively in terms of static articulatory configurations such as vocal tract length, articulator size, and length and thickness of the vocal folds, greater commonality could be in the dynamics of the articulatory trajectories, which is constrained by physical laws. The dynamic nature of continuous speech [55,56,57,58] means that it is intrinsically difficult to find the optimal time points at which discrete features can be extracted from the continuous articulatory or acoustic trajectories. Second, there is evidence that continuous articulatory (and the resulting acoustic) movements are divided into syllable-sized unidirectional target approximation movements [59,60,61] or gestures [62]. This suggests that processing syllable-sized acoustic signals could be an effective perceptual strategy to capture the full details of all the relevant information about contrastive phonetic units such as consonants, vowels, and tones. Finally, a seemingly trivial but in fact critical reason is that, if detailed acoustic signals are all available to the auditory system, should perception throw away any part of the signal that is potentially helpful? The answer to this question would be no, according to the data processing theorem, also known as data processing inequality [63]. This is an information theoretic concept that states that the information content of a signal cannot be increased via data processing:If X → Y → Z (Markov chain), then I (X; Y) ≥ I (X; Z), I (Y; Z) ≥ I (X; Z).(1)
Equality if I (X; Y|Z) = 0.
where X, Y and Z form a Markov Chain (a stochastic process consisting of a sequence of events, where the probability of each event depends on the state of the previous event). X is the input, Z is the processed output, and Y is the only path to convert X to Z. What this says is that whenever data is processed, some information is lost. In the best-case scenario, the equality could still largely hold when some information is lost but no processing can increase the amount of original information. In general, the more data processing, the greater the information loss. An extraction of intermediate features before phonetic recognition, therefore, would necessarily involve more processing than direct recognition of phonetic categories from raw acoustic signals.

There have already been some theories that favor relative direct or holistic speech perception. Direct realism [64], for example, argues that speech perception involves direct recognition of articulatory gestures, without the intermediatory of explicit representation of auditory features. However, because gestures are also sub-phonemic (Figure 1), an extra step is still needed to convert them to syllables and words. The exemplar theories [65,66,67] also postulate that in both production and perception, information about particular instances (episodic information) as a whole is stored. Categorization of an input is accomplished by comparison with all remembered instances of each category. It is suggested that people use already-encountered memories to determine categorization, rather than creating an additional abstract summary of representations. In exemplar models of phonology, phonological structures, including syllables, segments and even sub-segmental features emerge from the phonetic properties of words or larger units [67,68,69], which implies that units larger than phonemes are processed as a whole. The exemplar theories, however, have not been highly specific on how exactly such recognition is achieved.

In fact, there is a general lack of step-by-step procedural account of any of the theoretical frameworks on speech perception that starts from the processing of continuous acoustic signals. Auditory theories have presented no demonstration of how exactly continuous acoustic signals are converted into auditory cues in the first phase of perception, and how these representations are translated into consonants, vowels, and tones. Motor theory has suggested that gestures can be detected from acoustic signals through analysis by synthesis [31], but this has not yet been tested in perception studies, and has remained only as a theoretical conjecture. What is needed is to go beyond purely theoretical discussion of what is logically plausible, and start to test computationally what may actually work. For this purpose, it is worth noting that computational speech recognition has been going on for decades, in research and development in speech technology. As a matter of fact, automatic speech recognition (ASR) has been one of the most successful areas in speech technologies [70,71]. Zhang et al. [72], for example, reported word-error-rates (WERs) as low as 1.4%/2.6%.

The state of the art in speech recognition, however, does not use intermediate feature extraction as a core technology. Instead, units like diphones or triphones are directly recognized from continuous acoustic signals [73,74,75,76,77]. There have been some attempts to make use of features in automatic speech recognition. For example, the landmark-based approach tries to extract distinctive features from around acoustic landmarks such as the onset or offset of consonant closure, which can then be used to make choices between candidate segments to be recognized [14,78,79,80]. In most cases, however, systems using landmarks or distinctive features are knowledge-based, and the detected features are used as one kind of feature added on top of other linguistic features and acoustic features to facilitate the recognition of phonemes [81,82]. In this kind of process, there is no test of the effectiveness of distinctive features relative to other features. Some other automatic speech recognition systems use acoustic properties around the landmarks, but without converting them to any featural representations [83,84,85,86]. What is more, those recognition systems still use phoneme as the basic unit, which implies that phoneme is the basic functional speech unit, and units under phonemes do not have to be categorical. There have also been systems that make some use of articulatory features. However, there is no system that we know of that performs phonetic recognition entirely based on articulatory gestures extracted from acoustic signals.

Feature-to-percept, therefore, is questionable as a general strategy of speech perception, especially given the possibility of direct phonetic perception as an alternative. There has not yet been any direct comparisons of the two strategies, however. In light of the data processing theorem in Equation (1), both strategies would involve data processing that may lead to information loss. In addition, a strategy that can generate better perceptual accuracy could be computationally too costly. Therefore, in this study, a set of modelling experiments are conducted to compare the two perceptual strategies, measured in terms of recognition accuracy and computational cost. As the very first such effort, the object of recognition is Mandarin tones, because they involve fewer acoustic dimensions than consonants and vowels, as explained next.

### 1.3. Tone Recognition: A Test Case

In languages like Mandarin, Yoruba, and Thai, words are distinguished from each other not only by consonants and vowels, but also by pitch patterns known as tones. Tone in these languages therefore serves a contrastive function like consonants and vowels. Syllables with the same CV structure can represent different words when the pitch profiles in the syllable vary. Although tonal contrasts are sometimes also accompanied by differences in consonants, vowels and voice quality, pitch patterns provide both sufficient and dominant cues for the identification of tones [59,87].

How to define tonal contrasts is a long-standing issue for tone language studies. In general, efforts have predominantly focused on establishing the best featural representation of tones, starting from Wang’s [88] binary tone features in the style of distinctive features of Jakobson et al. [7]. Later development has moved away from simple binary features. For East Asian languages, a broadly accepted practice is to use a five-level system [89] which assumes that five discrete levels are sufficient to distinguish all the tones of many languages. Also different from the classical feature theory, the five-level system represents pitch changes over time by denoting each tone with two temporal points. The four tones of Mandarin, for example, can be represented as 55—tone 1, 35—Tone 2, 214—Tone 3, and 51—Tone 4, where a greater number indicates a higher pitch. Two points per tone is also widely used for African tone languages [90,91,92], although for those languages usually only up to three pitch levels, High, Mid, Low, are used. Morén and Zsiga [93] and Zsiga and Nitisaroj [94] even claimed that for Thai, only one target point per tone is needed for connected speech. There has also been a long-standing debate over whether pitch level alone is sufficient to represent all tones, or slope and contour specifications are also needed as part of the representation [4,5]. There are also alternative schemes that try to represent tone contours, such as the T value method, LZ value method [95,96,97], etc., but they also focus on abstracting the pitch contours into several discrete levels.

Under the feature-to-percept assumption, the two-point + five-level tone representation would mean that, to perceive a tone, listeners need to first determine if the pitch level is any of the five levels at each of the two temporal locations, so as to derive at a representation in the form of, e.g., 55, 35, 21 or 51. Those representations would then lead to the recognition of the tones. In such a recognition process, the key is to first detect discrete pitch levels at specific temporal locations before tone recognition. A conceivable difficulty with such tone feature detection is the well-known extensive amount of contextual variability. For example, due to inertia, much of the pitch contours of a tone varies heavily with the preceding tone, and it is only near the end of the syllable that the underlying tonal targets are best approached [98,99]. This would make tone level detection hard, at least for the first of the two temporal locations.

An alternative to the feature-to-tone scheme, based on the direct phonetic perception hypothesis, is to process continuous *f*_0_ contour of each tone-carrying syllable as a whole without derivation of intermediate featural representations. The plausibility of holistic tone processing can be seen in the success of tone recognition in speech technology. The state-of-the-art automatic tone recognition can be as accurate as 94.5% on continuous speech [100], with no extraction of intermediate tonal features. In fact, the current trend is to process as many sources of raw acoustics as possible, including many non-*f*_0_ dimensions in complex models [100,101]. This suggests that maximization of signal processing rather than isolation of distinctive cues may be the key to tone recognition.

Tone recognition would therefore serve as a test case for comparing direct and two-phase perception. But the speech-technology-oriented approach of using as many acoustic properties as possible makes it hard to isolate the key differences between the two approaches. Given that *f*_0_ alone is sufficient to convey most tonal contrasts in perception as mentioned above, in this study we will use a computational tone recognition task that processes raw *f*_0_ contours in connected Mandarin speech, with the aim to test if the perception of phonetic categories is more likely a two-phase feature-to-percept process or a single-phase direct acoustic decoding process. We will apply two machine learning algorithms to process Mandarin tones from syllable-sized *f*_0_ contours extracted from a connected speech corpus in ways that parallel different tone perception hypotheses, including direct perception, pitch level extraction, pitch profile features, and underlying articulatory targets.

## 2. Methods and Materials

The overall method is to use computational models to simulate tone perception as a semi-automatic recognition task by training them with syllable-sized *f*_0_ contours in connected speech. The perception strategies under comparison are simulated by different ways of processing the raw *f*_0_ contours, and the efficacy of each strategy is estimated in terms of recognition rate.

### 2.1. Recognition Models

Two recognition models are applied to recognize Mandarin tones. One is a supervised model, Support Vector Machine (SVM), which can be used to simulate conditions where learners already know the tone inventory of the language. As we only have *f*_0_ values as input, there is no need to use very complex models to train the data. The other model is an unsupervised model, Self-Organizing Map (SOM), which can be used to simulate conditions where learners have no knowledge of the tonal inventory in the language.

As shown in the experimental results to be reported later, the recognition rates achieved by the SOM model were much lower than those achieved by the SVM model, despite the promising results reported previously [102]. In addition, although SOM can simulate clustering of patterned data like *f*_0_ trajectories, it is difficult to simulate the extraction of abstract features. Therefore, SOM is applied only in the tone recognition experiments based on pitch contours (full *f*_0_ contours and pitch level detection).

#### 2.1.1. Support Vector Machine (SVM)

SVM is a supervised machine learning model developed for binary classification tasks. An SVM model is a representation of the examples as points in space, mapped so that the examples of the separate categories are divided by a clear hyperplane or gap that is as wide as possible. New examples are then mapped into that same space and predicted to belong to a category based on which side of the gap they fall. Visually, *f*_0_ contours of each tone may consist of different patterns and so can be mapped in different spaces as a whole in the training phase. During the testing phase, every sampled contour will get a probability of each tone category and be predicted as a certain tone. Then, we could get an average accuracy of each tone. In the application of SVM, all the training samples and testing samples are converted to D-dimensional vectors and labelled with +1 or −1. A simple functional margin can be: $f\left(x\right)=sign\left(W^{t}x+b\right)$. If the label is +1, $W^{t}x+b$ is expected to be larger than 0, otherwise it is smaller than 0. The weight W is a combination of a subset of the training examples and shows how each dimension of the vectors is used in the classification process. The vectors in this subset are called support vectors. In our experiment, one *f*_0_ contour is one sample consisting of 30 sample points, and treated as a 30-dimention vector. This is done with the LibSVM tool [103] with RBF kernel. It generalizes the binary classification to a n-class classifier that splits the task into n(n−1)/2 binary tasks and the solutions are combined by a voting strategy [104]. Five-fold cross-validations were applied on the training set automatically and randomly during training to optimize the model, and the classification accuracy of the testing set will be shown in the Results section to compare the performance of each model. The training and testing set will be introduced later in Section 2.2.

#### 2.1.2. Self-Organizing Map (SOM)

In contrast to SVM, SOM is an unsupervised machine learning algorithm that projects high-dimensional input space onto a discrete lower dimensional array of topologically ordered processing units. During this training process, the SOM model compresses the information while keeping the geometric relationships among input data. In the tone recognition task based on full *f*_0_ contours, the networks were designed to contain 100 units/prototypes, and all the *f*_0_ contours were put into the training model. After many iterations, each contour tends to approximate a certain unit and all the *f*_0_ contours are finally mapped onto the 10 × 10 prototypes. Observing the trained units, we could see that neighboured units are gradually varied and the clusters of units share similar characteristics based on *f*_0_ contours.

After training, every unit will have a tone property calculated by a firing frequency matrix. A unit with the probability of 68% or above for a tone is considered as categorized as that tone. During the testing phase, each *f*_0_ contour in the testing data was mapped onto a unit which means this contour was recognized as that tone. This categorization process is done with the “kohonen-package” in R [105].

For any highly abstract features to work, one of the first critical steps is to extract them from observations through identification and naming. This is not a trivial task, and its effectiveness can be shown only in terms of the ultimate rate of recognition of the phonetic category. For the five-level tone representation system and the two-level distinctive feature system mentioned earlier, pitch levels can be detected or recognized using SVM or SOM from *f*_0_ contours and then transformed into tone by simple mapping. For more abstract features like pitch profile features and underlying articulatory targets, features need to be extracted first in a particular model and then put into the tone recognition system (SVM).

### 2.2. Material

The data were syllable-sized *f*_0_ contours produced by four female and four male Mandarin speakers [98]. Each token is a 30 equidistant (hence time-normalized) discrete point vector taken from either the first or second syllable of a disyllabic tone sequence in the middle position of a carrier sentence. There was no differentiation of the tokens from the first and second syllables, leaving the information of syllable position in word/phrase unrepresented. Two frequency scales were used to represent *f*_0_, Hertz and semitones. The latter was converted from Hertz with the following equation:(2)$$\mathrm{semitone}=log_{2}\left(f_{0}\right)\times 12$$
where the reference *f*_0_ is assumed to be 1 Hz for all speakers. Note that this kind of raw data (i.e., without applying a normalization scheme such as Z-score transformation, c.f., [106]) leave most of the individual differences in pitch height intact, particularly between the female and male speakers, as can be seen in the plots of *f*_0_ contours in Figure 2 and Figure 3.

There were a total of 1408 tokens of Tone 1 (high-level), 1408 tokens of Tone 2 (rising), 1232 tokens of Tone 3 (low), and 1408 tokens of Tone 4 (falling). The fewer tokens of Tone 3 were because those of the first syllable followed by another Tone 3 were excluded to circumvent the problem of the well-known tone sandhi rule, which turns the first Tone 3 to be similar [89,98] though not identical [107] to Tone 2. The whole dataset was then divided into a training subset and a testing subset randomly, with a ratio of 2:1.

### 2.3. Overall Design

Five modelling experiments were set up to compare the efficacy of different tone recognition schemes, both cross-gender and cross-speaker. The first experiment used raw *f*_0_ contours of the four Mandarin tones to train an SVM model and an SOM model, respectively, which were then used to classify the tone categories. The second experiment, pitch level detection, again used raw *f*_0_ contours to train a SVM model and a SOM model, respectively. But this time, the models were used to detect pitch levels, which were then mapped to tone categories. In the two subsequent experiments, different tonal features were extracted from the raw *f*_0_ contours, and were then used to train the SVM models. The extracted features were then used to recognize the tones. The extracted tonal features were ordered from the most to the least adherent to the feature-to-percept paradigm:Pitch height (2 levels and 5 levels);*f*_0_ profile (slope etc.);Underlying pitch targets (quantitative target approximation (qTA) parameters).

## 3. Results

### 3.1. Experiment 1—Full f_0_ Contour

In the first experiment, the raw *f*_0_ contours were used to both train the tone recognition model and test the performance. As shown in Table 1, with raw *f*_0_ data, tone recognition rates based on SVM model were very high. In the mixed-gender condition, the recognition rates were 97.4% for contours in semitones and 86.3% for contours in Hertz. In the male-only condition, the recognition rate reached 99.1% for contours in semitones. In the female-only condition, the recognition rate was 96.6%. The much lower recognition rates for contours in Hz is not surprising, as the logarithmic conversion in calculating semitones has effectively normalized the vertical span of the pitch range, with only individual differences in pitch height still retained. The performances of the SOM model are lower than that of SVM, but even the rates in the mixed-gender condition were all above 70%. In later experiments, we will only focus on mixed-gender conditions.

Table 2 is the tone confusion matrix of *f*_0_ contours in semitones. The performance of the tone classification is similar to the human tone recognition reported by McLoughlin et al. [108] shown in Table 3. The corpus they used has a context-free carrier sentence structure that is similar to that used in the present study. Similar to the results in Table 2, their recognition rate is the highest for Tone 3 and lowest for Tone 4.

### 3.2. Experiments 2–3: Pitch Level Representation

In this experiment, we abstracted the *f*_0_ contours into a two-position height representation. We tested both a two-level (distinctive-feature style) and a five-level abstraction that would correspond to two popular featural representations of tones [88,89].

#### 3.2.1. Experiment 2—Distinctive Feature Style (Two Level) Representation

In a distinctive feature system, Mandarin tones can be represented by two levels: high and low. The four lexical tones of Mandarin can be represented as 11—Tone 1, 01—Tone 2, 00—Tone 3, and 10—Tone 4. ‘1’ means high and ‘0’ means low. In our implementation of this featural representation system, each *f*_0_ contour was split into two halves and each was labelled high or low. The first 15 points of the contours were labelled as 1—Tone 1, 0—Tone 2, 0—Tone 3, 1—Tone 4, and the later 15 points are labelled as 1—Tone 1, 1—Tone 2, 0—Tone 3, 0—Tone 4. Two sub-experiments were conducted. One is training and testing the two halves separately, and the other is training and testing the two halves together. The models used were SVM and SOM. In the first sub-experiment, after the classification, the results of the two halves are combined and checked.

Table 4 shows tone recognition rates of this experiment. The rates are 93.67% and 92.90% for the separate and together sub-experiments, respectively, based on the SVM model. Assuming that distinctive features of tones are unknown knowledge until after learning, SOM is more comparable to human tone perception than SVM. The recognition rates are 80.59% and 82.82% for the separate and together sub-experiments, respectively.

#### 3.2.2. Experiment 3—Five-Level Representation

In a five-level, hence non-binary, pitch level representation, the four lexical tones of Mandarin can be represented as 55—Tone 1, 35—Tone 2, 21—Tone 3, and 53—Tone 4, as shown in Figure 4. In the featural representation of this system, each *f*_0_ contour is again split into two halves, each consisting of 15 points. Again, two sub-experiments were conducted. One is training and testing the two halves separately, and the other is training and testing them together. The models used were SVM and SOM. In the first sub-experiment, after the classification, the results of the two halves are combined and checked.

Table 5 shows recognition rates of both SVM and SOM. For SVM, when the two halves of the *f*_0_ contours are trained separately, the recognition rate reached around 90%. When the two halves are trained together, the rate dropped to just above 80%. The results of SOM are even lower than SVM. The recognition rates are 58.9% and 43.7% for the separate and together sub-experiments, respectively.

### 3.3. Experiment 4—f_0_ Profile Representation

Besides the discrete representations tested so far, there are also schemes that use mathematical functions to represent *f*_0_ profiles with parameters with continuous values. A recent study explored fitting the tone contours with two mathematical functions, parabola and broken-line (BL) and concluded that three of the cues obtained in the parabola fitting were broadly effective: mean *f*_0_, slope, and curve [109]. In this experiment we tested the effectiveness of fitting both functions to the *f*_0_ contours in the test corpus in the least-squared sense. The expression of the parabola is as follows:(3)$$f\left(t\right)\approx c_{0}+c_{1}\left(t-\frac{1}{2}\right)+c_{2}\left[{\left(t-\frac{1}{2}\right)}^{2}-1/12\right]$$

The expression of BL is as follows:(4)$$f\left(t\right)\approx \{\begin{array}{c} a_{1}+b_{1}t,\;\;t<d\\ a_{2}+b_{2}t,\;\;t\geq d \end{array}\;\left(d\;\mathrm{is}\;\mathrm{the}\;\mathrm{position}\;\mathrm{of}\;\mathrm{breakpoint}\right)$$

The features we used for testing were the top five pairs of features reported in Tupper et al. [109] for maximizing classification accuracy, as follows:Slope: $c_{1}$ in the parabola fit;Curve: $c_{2}$ in the parabola fit, which is one half the second derivative of the fitted *f*_0_ contour;Onglide: difference between *f*_0_ at contour onset and breakpoint in the BL fit;Offglide: difference between *f*_0_ at breakpoint and contour offset in the BL fit;Overall: difference between *f*_0_ at contour onset and offset in BL fit.

The features extracted from *f*_0_ contours are trained by the SVM model. Table 6 shows the recognition rates of the top five pairs of features used in Tupper et al. [109]. The best results for mixed genders are 92.3% in semitones and 89.3% in hertz, both of which are based on slope + curve.

### 3.4. Experiment 5—qTA Articulatory Feature Extraction

qTA is another mathematic function also capable of representing tonal contours [59]. The model is based on the assumption that speech articulation is a mechanical process of target approximation that can be simulated by a critically damped spring-mass system (similar to the command-response model [110] and the task dynamic model [111]). The model can be fitted to not only tonal contours in continuous speech, but also intonational contours carrying multiple communicative functions [107]. QTA’s ability to simulate the articulatory process of generating tonal contours makes it an ideal model to test the motor theory, according to which speech perception is a process of detecting the articulatory gestures that generate the speech signals [31] through analysis-by-synthesis [112]. Analysis-by-synthesis is a process of analysing a signal by reproducing it, and it has been successfully applied in previous modelling works with qTA [59,107,113]. In this experiment, we used analysis-by-synthesis to fit qTA to the tonal contours in the same corpus used in the other experiments.

qTA assumes that the *f*_0_ contour of each syllable is generated with a single pitch target, defined by the linear function,
(5)x(t)=mt+b
where m (in st/s) and b (in st) denote the slope and offset of the underlying pitch target, respectively. The surface $f_{0}$ is modeled as the system response driven by the pitch target,
(6)$$f_{0}\;\left(t\right)=\left(mt+b\right)+\left(c_{0}+c_{1}\;t+\cdots +c_{N-1}t^{N-1}\right)e^{-t/\tau }$$
where the time constant τ (in s) represents the strength of the target approximation movement.

The values of m, b, and τ (referred to as qTA parameters) can be determined by fitting the original pitch contour in the least-squares sense. We used Target Optimizer [114] to extract qTA parameters. The Target Optimizer internally converts the $f_{0}$ samples from Hz scale to semitone scale and normalizes them by subtracting the mean values of the whole utterance. The three estimated qTA parameters were then used as input to a tone recognizer.

In qTA, the offset $f_{0}$ of the preceding syllable is transferred to the current syllable to become its onset $f_{0}$ to simulate the effect of inertia. Therefore, the onset *f*_0_ of a syllable is expected to be potentially relevant to tone recognition, as it carries contextual tonal information. In the second training condition, therefore, this onset $f_{0}$ was added to the qTA parameters to form a four-dimensional input feature for each syllable. These four-dimensional features are then used as input to the SVM model for tone recognition.

Table 7 shows the performance of qTA features from the two training conditions. With the three-dimensional features, the recognition accuracy was 90.7%. With the four-dimensional features, which included the *f*_0_ onset parameter, the accuracy increased to 97.1%.

## 4. Discussion

In the five experiments, we tested whether direct phonetic perception or feature-to-percept is a more likely mechanism of speech perception. All the tests were done by applying the SVM and/or SOM model with either full *f*_0_ contours or various extracted *f*_0_ features as the training and testing data. Except for qTA (due to model-internal setting), all the models were tested with *f*_0_ in both Hz and semitone scales. The performance of the models was assessed in terms of tone recognition rate. In all the experiments the recognition was consistently better for the SVM model than for the SOM model, and better with the semitone scale than the Hz scale. To make a fair comparison of the all the models, a summary of the best performances in all five experiments based on SVM in semitones is shown Figure 5. As can be seen, the highest recognition rate, 97.4%, was achieved in the full *f*_0_ contour condition in Experiment 1. With pitch level features extracted in Experiments 2–3, recognition rates of 93.7% and 90.3% were achieved for the two-level and five-level conditions, respectively. These are fairly good, but are well below the top recognition rate in Experiment 1. Experiment 4 tested two mathematical (parabola and broken-line) representations of *f*_0_ profiles, which, unlike the discrete pitch level features in Experiments 2–3, have continuous values. The highest recognition rate of 92.3% was achieved for the combination of slope and curve. This is very close to the best recognition rate of 93.7% with the two-level condition in Experiment 2. Another continuous parametric representation tested in Experiment 5, namely, qTA parameters based on [59], achieved a very high recognition rate of 97.1% when initial *f*_0_ was included as a fourth parameter, which is almost as high as the benchmark of 97.4% in Experiment 1. Without the initial *f*_0_, however, the recognition rate was only 90.7%. It is worth noting, however, that the initial *f*_0_ is actually included in the full *f*_0_ contour in Experiment 1, as it is just the first *f*_0_ point in an *f*_0_ contour.

The fairly high tone recognition rates from the best performances in all the five experiments are rather remarkable, given that the *f*_0_ contours used were extracted from fluent speech [51] in multiple tonal contexts and two different syllable positions, yet no contextual or positional information was provided during either training or testing, contrary to the common practice of including tonal context as an input feature in speech technology [101,115,116]. This means that, despite the extensive variability, tones produced in contexts by multiple speakers of both genders are still sufficiently distinct to allow a pattern recognition model (SVM) to accurately identify the tonal categories based on syllable-sized *f*_0_ contours alone. In other words, once implemented as trainable systems, most theory-motivated schemes may be able to perform phonetic recognition to some extent, though with varying levels of success. Thus, the acoustic variability that has prompted much of the early theoretical debates [30,117] does not seem to pose an impenetrable barrier. Instead, there seems to be plenty of consistency underneath the apparent variability for any recognition scheme to capture.

On the other hand, it is still the case that the tone recognition rates achieved by most of the feature extraction schemes in Experiments 2–5 were lower than that of the full *f*_0_ contour baseline in Experiment 1. Only the qTA + initial *f*_0_ scheme nearly matched the full *f*_0_ contour performance. Therefore, for both the qTA + initial *f*_0_ condition and the other feature extraction schemes, a further question is whether the extra processing required by the extraction of the intermediate features is cost-effective when compared to direct processing of full *f*_0_ contours. One way to compare the cost-effectiveness of different tone recognition schemes is to calculate their time complexity [118] in addition to their recognition rates. Time complexity is the amount of time needed to run an algorithm on a computer, as a function of the size of the input. It measures the time taken to execute each statement of the code in an algorithm and gives information about the variation in execution time when the number of operations changes in an algorithm. It is difficult to compute this function exactly, so it is commonly defined in terms of an asymptotic behaviour of the complexity. Time complexity is expressed with the big O notation, O[n], where n is the size of the input data and O is the order of the relation between n and the number of operations performed. Taking the SVM model as an example, the time complexity of the SVM model at testing phase is a function that involves a loop within a loop, which is O(d)×O(m)=O(d×m), where d is the number of dimensions of input data and m is the number of categories. When two models, A and B, are compared, if A is better than or equal to B in performance, and has a lower time complexity, A can be said to be a better model than B. If A and B differ clearly in performance, but has a lower time complexity, its performance needs to be balanced against time complexity when deciding which model is better. Table 8 shows the time complexity of all the tone recognition schemes tested in Experiments 1–5.

As can be seen, most feature extraction schemes have greater time complexity than the baseline full *f*_0_ contour scheme. The only feature extraction scheme with lower time complexity is the two-level condition. However, its tone recognition accuracy is 3.7% lower than the full *f*_0_ contour condition, as shown in Figure 5. Therefore, its reduced time complexity was not beneficial. In addition, as found in Chen and Xu [119], the time complexity of full *f*_0_ contour scheme does not need to be as high as in Experiment 1, because the temporal resolution of *f*_0_ contours could be greatly reduced without lowering tone recognition accuracy. When the number of *f*_0_ points were reduced to as few as 3, the tone recognition rate was still 95.7%, only a 1.7% drop from the 30-point condition. Therefore, compared to 3 points per contour, the two-level feature extraction would even lose its advantage in time complexity. Overall, therefore, there is no advantage in cost-effectiveness in any of the features extraction schemes over the full *f*_0_ contour scheme.

Worth particular mentioning is the qTA scheme tested in Experiment 5, as it served as a test case for the motor theory of speech perception [31]. The theory assumes that speech perception is a process of recovering articulatory gestures, and the recovery is done by listeners using their own articulatory system to perform analysis-by-synthesis. However, analysis-by-synthesis is a time-consuming process of testing numerous candidate model parameters until an optimal fit is found. As shown in Table 8, the qTA scheme has the greatest time complexity of all the feature extraction schemes. Although its tone recognition accuracy was nearly as high as that of full *f*_0_ contours benchmark when initial *f*_0_ was included as the fourth parameter, one may have to wonder why speech perception would develop such an effortful strategy when direct processing of raw *f*_0_ contours can already achieve top performance at a much lower computational cost.

An implication of the experiments in the present study is that listeners’ sensitivity to certain feature-like properties, such as *f*_0_ slope [5], height [4] or alignment of turning point [120,121] does not necessarily mean that those properties are separately extracted during perception. Rather, the sensitivity patterns are what can be observed when various acoustic dimensions are independently manipulated under laboratory conditions. They do not necessarily tell us how speech perception operates. The step-by-step modelling simulations conducted in the current study demonstrate that there may be no need to focus on any specific features. Instead, the process of recognition training allows the perception system to learn how to make use of all the relevant phonetic properties, both major and minor, to achieve optimal phonetic recognition. The dynamic learning operation may in fact reflect how phonetic categories are developed in the first place. That is, speech production and perception probably form a reciprocal feedback loop that guarantees that articulation generates sufficiently distinct cues that perception can make use of during decoding. As a result, those articulatory gestures that can produce the greatest number of effective phonetic cues would tend to be retained in a language. At the same time, perceptual strategies would tend to be those that can make the fullest, and the most economical, use of all the available acoustic cues from detailed acoustic signals.

Finally, a few caveats and clarifications are in order. First, the tone recognition rates obtained in the present study may not directly reflect perception efficacy in real life. On the one hand, they could be too high because the *f*_0_ contours tested here did not contain some of the known adverse effects like consonantal perturbation of *f*_0_ [122,123], intonational confounds [99,124], etc. On the other hand, they could also be too low because not all tonal information is carried by *f*_0_. Listeners are also known to make use of other cues such as duration, intensity, voice quality, etc. [100,101]. Second, there is a need to make a distinction between data transformation and feature extraction. Conversion of *f*_0_ from Hz to semitones and from waveform to MFCC are examples of data transformation. Both have been shown to be beneficial [125,126], and are likely equivalent to certain signal processing performed by the human auditory system. They are therefore different from the feature extraction schemes tested in the present study. In addition, there is another kind of data processing that has been highly recommended [106,127], namely, speaker/data normalization schemes in the frequency dimension such as Z-score transformation (rather than in the temporal dimension). The justification is based on the need to handle variability within and especially across speakers. The difficulty from an operational perspective is that Z-score transformation is based on the total pitch range of multiple speakers in previously processed data. However, Z-score would be hard to compute when processing data from a new speaker, which happens frequently in real life. Furthermore, the present results have shown that, once an operational model is developed, explicit speaker normalization is not really needed, as the training process is already one of learning to handle variability, and the results showed that all models were capable of resolving this problem to various extends. Finally, the present findings do not suggest that a data representation of up to 30 points per syllable is necessary for tone recognition from continuous speech. As mentioned earlier, in a separate study [119] we found that just three *f*_0_ points (taken from the beginning, middle and end of a syllable) are enough for a model equivalent to the full contour model in Experiment 1 to achieve a tone recognition rate close to that of 30 *f*_0_ points, so long as the data points are in the original *f*_0_ values rather than discretized pitch height bands. The study also found that discretization of continuous acoustic signal into categorical values (equivalent to reduction of frequency resolution), which is prototypical of featural representations, is the most likely to adversely affect tone recognition. In other words, a temporal resolution of up to 30 samples per syllable as tested in the present study is not always needed, and may in fact be excessive when time complexity is taken into consideration, whereas high precision of data representation, which is exactly the opposite of featural representation, may be the most important guarantee of effective speech perception.

## 5. Conclusions

We have used tone recognition as a test case for a re-examination of the widely assumed feature-to-percept assumption about speech perception. Through computational simulation of tone recognition that applied step-by-step modelling procedures, we put various theoretical accounts of speech perception to test by making all of them process continuous acoustic signals. The results show that syllable-sized *f*_0_ contours can be used to directly train pattern recognition models to achieve high tone recognition rates, without extracting intermediate features. In comparison, extracting discrete pitch levels or continuous profile features from the original *f*_0_ contours resulted in reduced rates of tone recognition. Furthermore, when articulatory-based qTA parameters were extracted through analysis-by-synthesis, an operation reminiscent of the motor theory of perception, the recognition rate approached that of original *f*_0_ contours only when syllable-initial *f*_0_ was used as an additional parameter. Finally, we showed through calculation of time complexity relative to model performance that all the feature extraction schemes are less cost effective than the full *f*_0_ contour condition. Based on these findings, we conclude that raw acoustic signal, after certain transformations such as semitone (or MFCC for segments) conversion, can be processed directly in speech perception to recognize phonetic categories. Therefore, feature detection, while useful for analysis and observational purposes, is unlikely to be the core mechanism of speech perception. While the present findings still cannot tell us how exactly speech perception works, they have at least presented a computational reason why real-life speech perception is unlikely a two-phase process, something that is hard to observe through behavioral or state-of-the-art neural investigations alone.

## Figures and Tables

**Figure 1 brainsci-12-00337-f001:**
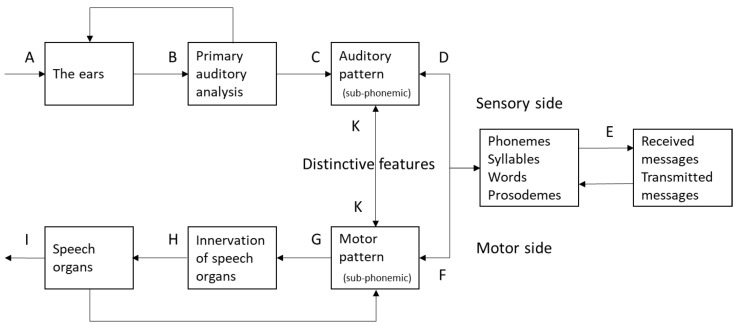
Hypothetical model of brain functions in speech perception and production (adapted from Fant, 1967 [53]).

**Figure 2 brainsci-12-00337-f002:**
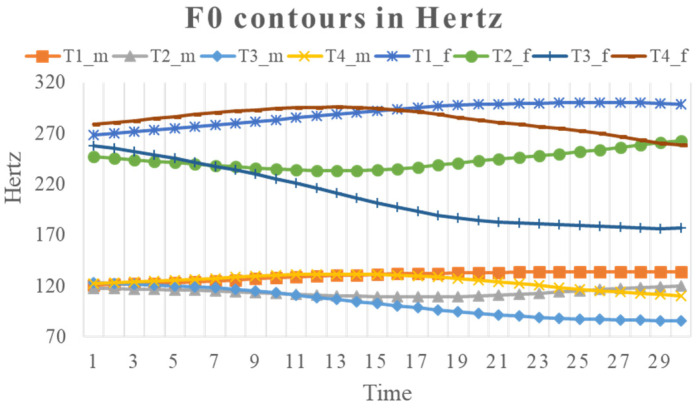
Mean time-normalized syllable-sized *f*_0_ contours of four Mandarin tones, averaged separately for female and male speakers.

**Figure 3 brainsci-12-00337-f003:**
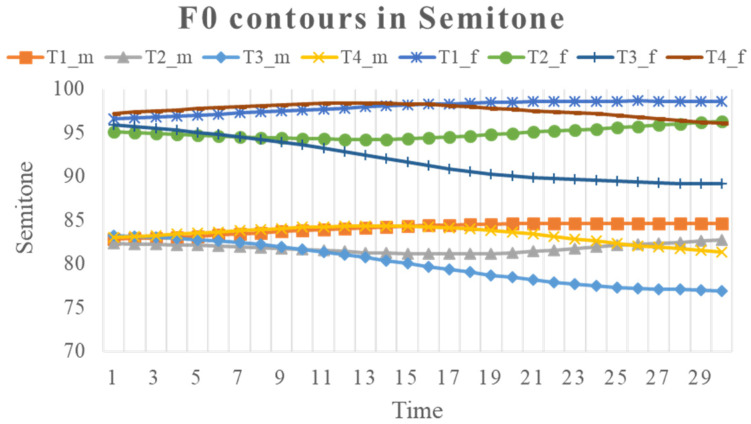
Mean time-normalized syllable-sized semitone contours of four Mandarin tones, averaged separately for female and male speakers.

**Figure 4 brainsci-12-00337-f004:**
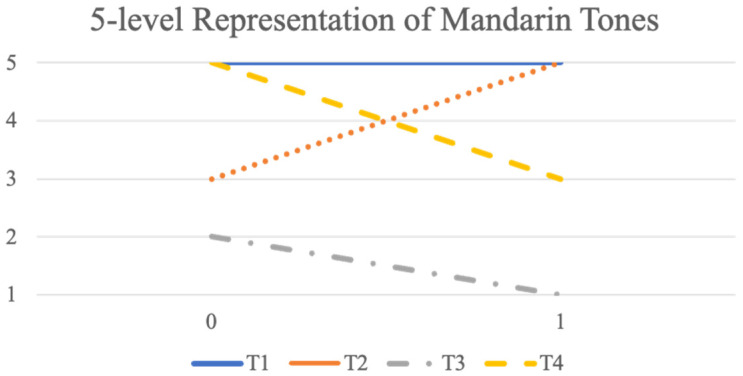
Five-level Representation of Mandarin Tones.

**Figure 5 brainsci-12-00337-f005:**
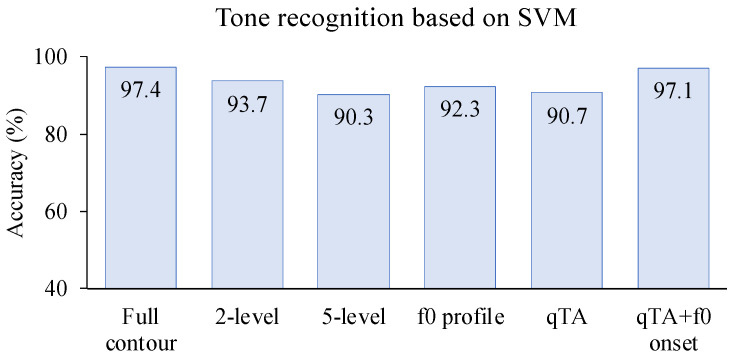
Summary of tone recognition rates based on SVM model for full *f*_0_ contour (Experiment 1), two-level feature (Experiment 2), five-level feature (Experiment 3), *f*_0_ profile (Experiment 4) and qTA and qTA + *f*_0_ onset (Experiments 5). SVM: Support Vector Machine.

**Table 1 brainsci-12-00337-t001:** Tone recognition rates using raw *f*_0_ contours based on SVM and SOM models.

	SVM		SOM	
	Hertz	Semitone	Hertz	Semitone
**Male**	96.0%	99.1%	89.7%	90.8%
**Female**	76.7%	96.6%	76.7%	77.6%
**All**	86.3%	97.4%	72.8%	72.0%

SVM: Support Vector Machine; SOM: Self-Organizing Map.

**Table 2 brainsci-12-00337-t002:** Tone confusion matrix using semitone of mixed-gender based on SVM.

	T1	T2	T3	T4
**T1**	98.2%	0.2%	0.5%	1.1%
**T2**	0.9%	96.6%	0.9%	1.6%
**T3**	0.3%	1.0%	98.4%	0.3%
**T4**	2.7%	0.0%	0.7%	96.6%

T means Tone here.

**Table 3 brainsci-12-00337-t003:** Tone confusion matrix context-free words in AWGN-corrupted spoken sentences [108].

	T1	T2	T3	T4
**T1**	95.68%	2.03%	2.08%	0.21%
**T2**	1.24%	97.92%	0.08%	0.76%
**T3**	0.35%	0.42%	99.02%	0.21%
**T4**	2.63%	1.72%	1.38%	94.27%

**Table 4 brainsci-12-00337-t004:** Tone recognition rates using two-level abstraction based on SVM and SOM models.

	Separate		Together	
	Hertz	Semitone	Hertz	Semitone
**SVM**	93.5%	93.7%	91.9%	92.9%
**SOM**	80.8%	80.6%	81.0%	82.8%

SVM: Support Vector Machine; SOM: Self-Organizing Map.

**Table 5 brainsci-12-00337-t005:** Tone recognition rates using five-level abstraction based on SVM and SOM models.

	Separate		Together	
	Hertz	Semitone	Hertz	Semitone
**SVM**	88.8%	90.3%	80.5%	84.1%
**SOM**	56.7%	58.9%	43.5%	43.7%

SVM: Support Vector Machine; SOM: Self-Organizing Map.

**Table 6 brainsci-12-00337-t006:** Tone recognition rates using *f*_0_ profile features based on SVM model.

	Herz	Semitone
**Slope + Curve**	89.3%	92.3%
**Curve + Overall**	85.9%	90.4%
**Slope + Onglide**	68.3%	66.7%
**Onglide + Offglide**	71.8%	75.7%
**Offglide + Overall**	70.4%	75.1%

**Table 7 brainsci-12-00337-t007:** Tone recognition rates using qTA (quantitative target approximation) features.

Features	Accuracy
**3-dim qTA parameters**	90.7%
**3-dim qTA parameters plus *f*_0_ onset value**	97.1%

**Table 8 brainsci-12-00337-t008:** Time complexity of different tone recognition schemes.

Scheme	Method	Size of Input	No. Steps	Time Complexity at Testing Phase
**Full *f*_0_ contour**	SVM	30 points	1	O(30 × 4)
**2 *f*_0_** **levels**	SVM × 2	15 points	3	O(15 × 2) × 2 + 1
	Matching	2 features		
**5 *f*_0_** **levels**	SVM × 2	15 points	3	O(15 × 5) × 2
	Matching	2 features		
** *f* _0_ ** **profile features**	Parabola/Broken Line	30 points	2	O(30^3^) + O(2 × 4)
	SVM	2 features		
**qTA features**	qTA Extraction	30 points	2	O(30^3^) + O(3 × 4)
	SVM	3 features		

SVM: Support Vector Machine. Qta: quantitative target approximation.

## Data Availability

The original corpus is available upon request from the third author. The data from the simulations are available upon request from the first author.

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
