# Peer review of "Computational Modelling of Tone Perception Based on Direct Processing of f0 Contours"

_brainsci, 2022, doi:10.3390/brainsci12030337_

Round 1

Reviewer 1 Report

Summary:

The manuscript first introduced speech perception theories from a perspective where an intermediate level of abstraction (acoustic or gestural) is needed to categorize phonemes. The authors argued for an alternative where the abstraction process is not needed and that the ‘raw’ data can be used to directly categorize speech signals. The authors presented results from a series of computation experiments where models were trained on ‘raw’ data to directly categorize speech or models were trained on ‘raw’ data to extract intermediate ‘features’ and then the ‘features’ are further used for categorization. The authors did a good job in setting up the argument and the approach is very interesting. I have a few comments to hopefully help improve the methods/interpretation.

  1. From my understanding, the authors argue that a ‘direct categorization’ theory should be considered because overall the ‘full contour’ model did better than the ‘intermediate abstraction’ models. However, I find the difference hard to interpret. Were there any cross-validation of the models done, or other measures such as permutation, such that we can get an error estimate on the model performances? Can it just be by chance due to how the training vs. testing set were set up to get these performance score?
  2. To me, it is not surprising that with 30 data point that a model can do better than with 2 data points, however, the difference may not be big enough to justify processing the full ‘raw’ data? The author further argues that the processing complexity is smaller in the 1-step strategy, hence the 1-step strategy should be the primary route. I wonder if the ‘processing complexity’ is sufficient for this argument. As the authors discussed in the introduction, the reasoning for feature abstraction may be for adverse conditions. I wonder if it needs to be shown that a 1-step strategy is still better when a less than ideal f0 can be extracted, for example, adding noise to the signal?

Some small comments:

  1. I’m not sure why the time complexity result is only presented in the discussion section. It seems it should be in result section. Line 571, it will also be beneficial to add in some interpretation for people who are not familiar with the time complexity measure. Such as, the larger the N, the more complex?
  2. Line 157-166. It is not super clear what the authors are trying to express. Because 2 step process lose more information therefore is less ideal? Also, the author may want to explain what a ‘Markov Sequence’ is for readers who are not familiar with it.

Reviewer 2 Report

The manuscript reports the results of a study that used computational modelling to explore whether it is possible to recognize phonetic units in connected speech (as the authors have defined that term), without an intermediate featural representation.  Questions like this could be important for theories of speech and language processing.  The authors believe that the approach used here might help answer questions about, in their words, "how exactly speech perception works".  I don't share that belief, but the approach may be useful for other, more applied problems, such as machine recognition of speech.  For that reason, the audience for this work is likely to be specialists involved in similar projects, rather than the larger community of speech scientists and psycholinguists.  As a related general comment, the Introduction seems overly long, largely because of the space devoted to a review of theories of speech perception that are mostly of historical interest (look at the dates on the cited papers).  It could be made shorter and more relevant if the emphasis were on contemporary views.  A more specific concern is the use of the phrase "continuous acoustic signals", which I think is misleading.  Most speech scientists would expect that phrase to refer to ongoing connected speech, in the form of a sound wave.  Here, it refers to a "syllable-sized" segment, reduced to a 30-point vector;  in other words, not continuous and not acoustic.  I understand why that was done, so the method is not the problem.  But please describe the input more accurately.

Round 2

Reviewer 1 Report

The revision looks good to me

Author Response

Point 1: The revision looks good to me.

Response 2: Thank you so much.